# The Prognostic Impact of Estimated Creatinine Clearance by Bioelectrical Impedance Analysis in Heart Failure: Comparison of Different eGFR Formulas

**DOI:** 10.3390/biomedicines9101307

**Published:** 2021-09-24

**Authors:** Pietro Scicchitano, Massimo Iacoviello, Andrea Passantino, Piero Guida, Micaela De Palo, Assunta Piscopo, Michele Gesualdo, Pasquale Caldarola, Francesco Massari

**Affiliations:** 1Cardiology Section, Hospital “F. Perinei”, 70022 Altamura, Italy; assuntapiscopo@alice.it (A.P.); michelegesualdo@libero.it (M.G.); franco_massari@libero.it (F.M.); 2Cardiology Unit, Department of Medical and Surgical Science, University of Foggia, 71122 Foggia, Italy; massimo.iacoviello@gmail.com; 3Division of Cardiology and Cardiac Rehabilitation, Scientific Clinical Institutes Maugeri, IRCCS Institute of Bari, 70124 Bari, Italy; andrea.passantino@icsmaugeri.it; 4Regional General Hospital “F. Miulli”, 70021 Acquaviva delle Fonti, Italy; pieroguida@libero.it; 5Cardiac Surgery Section, Policlinico University Hospital, 70124 Bari, Italy; micaela.depalo85@gmail.com; 6Cardiology Section, Hospital “S. Paolo”, 70123 Bari, Italy; pascald1506@gmail.com

**Keywords:** acute heart failure, chronic heart failure, BIA, BNP, clearance creatinine, prognosis

## Abstract

The estimation of glomerular filtration rate (eGFR) provides prognostic information in patients with heart failure (HF). Bioelectrical impedance analysis may calculate eGFR (Donadio formula). The aim of this study was to evaluate the impact of the Donadio formula in predicting all-cause mortality in patients with HF as compared to Cockroft-Gault, MDRD-4 (Modification of Diet in renal Disease Study), and Chronic Kidney Disease Epidemiology Collaboration (CKD-EPI) formulas. Four-hundred thirty-six subjects with HF (52% men; mean age 75 ± 11 years; 42% acute HF) were enrolled. Ninety-two patients (21%) died during the follow-up (median 463 days, IQR 287–669). The area under the receiver operator characteristic curve for eGFR, as estimated by Cockroft-Gault formula (AUC = 0.75), was significantly higher than those derived from Donadio (AUC = 0.72), MDRD-4 (AUC = 0.68), and CKD-EPI (AUC = 0.71) formulas. At multivariate analysis, all eGFR formulas were independent predictors of death; 1 mL/min/1.73 m^2^ increase in eGFR—as measured by Cockroft-Gault, Donadio, MDRD-4, and CKD-EPI formulas—provided a 2.6%, 1.5%, 1.2%, and 1.6% increase, respectively, in mortality rate. Conclusions. eGFR, as calculated with the Donadio formula, was an independent predictor of mortality in patients with HF as well as the measurements derived from MDRD4 and CKD-EPI formulas, but less accurate than Cockroft-Gault.

## 1. Introduction

The assessment of glomerular filtration rate (GFR) is mandatory in the management of patients with heart failure (HF). About 23% of patients with HF show worsening renal function (WRF) as a consequence of the impairment in kidney perfusion, pharmacological treatments, and neuro-hormonal activation [1,2,3]. The literature has reported prevalence in WRF in acute decompensated HF patients ranging from 25% to 40% [4].

The worsening in kidney function in both acute (AHF) and chronic (CHF) patients is related to poor outcomes [1]. It is associated with a two-fold increase in all-cause mortality risk [1].

No matter the type of HF [5,6], WRF is related to a 2-3-fold increase in adverse in-hospital outcomes and 1.5-fold increase in 1-year mortality [5].

Therefore, the need for monitoring renal function is one of the mainstays in the general management of patients with both AHF and CHF. Several formulas and equations have been created to assess and monitor the performance of the kidneys during hospital stay for HF: Cockroft-Gault, Modification of Diet in Renal Disease Study (MDRD-4), and the Chronic Kidney Disease Epidemiology Collaboration (CKD-EPI) are the best representative formulas in this setting [7,8,9]. The literature has provided different results about the prognostic value of each formula in HF patients [10,11,12,13,14]. According to Zamora et al. [14], CG has the most accurate risk predictive skill compared to MDRD and CKD-EPI in HF patients. McAlister et al. [15] compared MDRD and CKD-EPI as predictors of mortality in HF patients, finding that CKD-EPI seemed more reliable than MDRD in predicting all-cause mortality at 3-year follow-up. Casado Cerrada et al. [16] demonstrated that CKD-EPI was able to better re-classify patients with AHF (net reclassification improvement [NRI]: 6.78%).

Indeed, the influence of body weight, muscular mass, and creatinine production may limit the use of creatinine-based formulas. The application of bioimpedance analysis (BIA) to the estimation of glomerular filtration rate (eGFR) may provide further insights to clinicians [17,18]. Donadio et al. [17,18] estimated GFR from body cell mass (BCM), as assessed by total body BIA. The value of BCM is strictly correlated with 24 urine creatinine excretion (UCr) [17]; therefore, it is possible to estimate GFR by combining BCM and plasma creatinine [17,18].

The aims of this study were to evaluate the prognostic value of eGFR as assessed by the Donadio formula, and to compare it to Cockroft-Gault, MDRD-4, and CKD-EPI formulas in HF patients.

## 2. Materials and Methods

### 2.1. Study Populations

This was a retrospective study based on a cohort of patients who were consecutively admitted to the Cardiology Unit of “F. Perinei” Hospital in Altamura (Bari, Italy), due to AHF or CHF, between January 2016 and November 2019.

The dataset included the clinical/anthropometric characteristics of the study population, blood biochemical data, BIA measurements, and pharmacological treatments. The left ventricular ejection fraction (LVEF) was calculated with Simpson’s method by means of echocardiography. BNP levels were assessed using a microparticle enzyme immunoassay (Architect, Abbott Park, IL, USA), while serum creatinine was measured with a Beckman Coulter AU 680 chemistry analyser (Beckman Coulter s.r.l. Via Roma, 108—Cassina Plaza 20060—Cassina De’ Pecchi Milan, Italy). All these measurements were routinely performed for all patients at admission to our ward.

Exclusion criteria consisted of myocarditis, pericarditis, pulmonary embolism, acute coronary syndrome, and recent cardiac surgery intervention.

The primary endpoint was all-cause death as ascertained from available medical records or National Death Records.

All patients gave their written informed consent to collect all the data related to their hospital stay. The performance of the study was in agreement with the Declaration of Helsinki. The study was approved by the local Institutional Review Board of the ASL BARI (protocol n. 0081801/CE—29 October 2015, study number: 4816).

### 2.2. Creatinine-Based Formulas for Estimating GFR

We calculated estimated GFR with the use of the following formulas: Donadio, Cockroft-Gault, MDRD-4, and CKD-EPI equations. GFR was expressed in mL/min/173 m^2^:(1)The Donadio formula was obtained by using tetrapolar impedance plethysmography, which emitted a single alternating sinusoidal current at 50 kHz (CardioEFG, Akern RJL Systems, Florence, Italy). The values of resistance and reactance were measured, and BCM was calculated according to the manufacturer’s equation. Urinary Creatinine Excretion (Ucr) estimation was obtained as follows: Ucr (mg/24 h) = BCM (Kg) × 30.2 + height (cm) × age (years) × 8.35 − 2222, while eGFR was calculated as: Ucr (mg)/plasma creatinine (mg/mL) × 1440 min [17,18];(2)The Cockroft-Gault formula: (140 − age) × (weight)/(72 × serum creatinine) × 0.85 (if female) [7];(3)The MDRD-4 formula: 186.3 × creatinine − 1.154 × age − 0.203 × 1.212 (if black) × 0.742 (if female) [8];(4)The CKD-EPI formula: male: 141 × minimum (creatinine/0.9, 1) − 0.411 × maximum (creatinine/0.9, 1) − 1.209 × 0.993Age × 1.159 (if black); female: 141 × minimum (creatinine/0.7, 1) − 0.329 × maximum (creatinine/0.7, 1) − 1.209 × 0.993Age × 1.018 × 1.159 (if black) [9].

### 2.3. Statistical Analysis

Normally distributed variables are reported as mean (standard deviation), and non-normally distributed continuous variables as median [25th–75th interquartile range (IQR)].

Because the distribution of BNP levels had a skewed distribution, a logarithm was used to obtain an optimal residual analysis. The comparisons between survivor and non-survivor groups were performed with a Student’s *t*-test or Mann-Whitney U-test where appropriate. Receiver-operating characteristic (ROC) curve analysis was performed to calculate the area under the curve (AUC) values, and the optimal cut-off values for mortality were calculated as the point of maximum sensitivity and specificity. The AUCs were compared using Hanley and McNeil tests. Survival was calculated using Kaplan-Meier analysis.

Univariate and multivariate analyses were performed to evaluate hazard ratios (HR) between eGFR formulas and all-cause mortality using Cox for proportional hazards. For multivariate analysis, we adjusted for variables considered to be of potential prognostic impact, such as ischemic heart disease, history of diabetes, atrial fibrillation, NYHA functional class, LVEF, serum sodium, BNP, hemoglobin, and HF status (AHF vs. CHF). After Pearson’s test between blood urea nitrogen and eGFR formulas (*r* > 0.60 for all), blood urea nitrogen dropped from multivariate Cox analysis because it was redundant. *p*-values below 0.05 were defined as statistically significant. The analyses were performed using STATA software, version 12 (StataCorp, College Station, TX, USA).

## 3. Results

There were 436 patients included in this study. Table 1 gathers the main characteristics of the patients.

The BCM was 24 ± 8 Kg, estimated Ucr was 1068 ± 381 mg and mean eGFR as derived from Donadio formula was 64 ± 35 mL/min/1.73 m^2^. The Cockroft-Gault, MDRD-4, and CKD-EPI formulas revealed the following mean eGFR values: 58 ± 27 mL/min/1.73 m^2^, 53 ± 27 mL/min/1.73 m^2^, and 53 ± 28 mL/min/1.73 m^2^, respectively.

In relation to the different formulas, the prevalence of renal insufficiency—as defined by eGFR < 60 mL/min/1.73 m^2^—was 53% with the Donadio formula, 57% with Cockroft-Gault formula, 69% with MDRD-4 formula, and 68% with CKD-EPI formula.

Ninety-two (21%) patients died during the follow-up (median 463 days, IQR 287–669). Patients with AHF showed a significantly higher mortality rate (31%) than those with CHF (13%) (*p* < 0.0001). ROC curves analysis tried to evaluate the role of eGFR formulas in predicting all-cause mortality risk and the optimal cut-off points to be considered for the best prediction (Table 2).

As reported in Figure 1, the ROC curve related to Cockroft-Gault formula was significantly higher than those derived from the other formulas.

The long-term survival Kaplan–Meier curves of the four formulas—categorized according to the different stages of kidney failure—are shown in Figure 2.

All the formulas are confirmed to be independent predictors of mortality in patients with HF. However, the curve from the Cockroft-Gault formula was more divergent than the others: chi-squared was 82 for Cockroft-Gault, 39 for Donadio, 39 for MDRD4, and 45 for CKD-EPI.

At univariate Cox proportional hazard analysis, all the eGFR formulas were predictors of death. After adjustment for HF covariates, all eGFR formulas confirmed to act as independent predictors of death. Specifically, each 1 mL/min/1.73 m^2^ point decrease in eGFR values as calculated by Cockroft-Gault, Donadio, MDRD4, and CKD-EPI formulas increased all-cause mortality risk by 2.6%, 1.5%, 1.2%, and 1.6%, respectively (Table 3 and Table 4).

## 4. Discussion

The prognostic value of eGFR in patients with HF represents a practical approach to the risk stratification of these patients by adopting findings from common biochemical data. Our study mainly demonstrated that four formulas—namely Donadio formula, CG, MDRD-4, and CKD-EPI—were able to assess the risk of all-cause mortality in HF patients. Specifically, our findings demonstrated that CG formula provided the best prognostic value in HF compared to the other three formulas by revealing an AUC value of 0.75 for a cut-off value < 50 mL/min/1.73 m^2^, with sensitivity equal to 82%, specificity 57%, PPV 33%, and NPV 92%.

This is the first study trying to evaluate the prognostic role of the Donadio formula in patients with HF. The major innovative characteristic of our research was to provide a comprehensive tool, bioimpedance analysis, able to evaluate the congestion status of patients with HF, the occurrence of cardio-renal syndrome by providing information about GFR calculation, and the stratification of their mortality risk. Such results may provide a useful bedside approach to the clinical practice in early identification of subjects with HF and in cases with a poor prognosis.

The poor performance of MDRD-4 in predicting adverse events in HF patients was already established in the literature [19], although better results were demonstrated in post-cardiac transplantation recipients [20] or patients with diabetes [21]. CKD-EPI provided better risk stratification than MDRD-4 in patients with HF after acute decompensation, although the AUC of the ROC curve did not overcome 0.64 in a study by Casado Cerrada et al. [16]. Further studies revealed the better prognostic value of CKD-EPI as compared to MDRD when evaluating the impact of GFR on the overall survival rate of patients with HF [15,22,23,24]. Indeed, the use of cystatin in CKD-EPI calculation provided a reason for the better performance of this equation in the setting of HF patients [25,26,27,28,29,30]. Our study pointed out that an CKD-EPI equation based on serum creatinine alone was able to predict all-cause mortality in patients with HF (AUC: 0.71); this prognostic value was maintained after multivariate Cox regression analysis. 

The CG formula was the best formula for stratifying the risk of death in HF patients as compared to MDRD and CKD-EPI [14,31,32,33]. The AUC of GFR, as estimated by CG, was higher than 0.80 in a cohort of 925 HF outpatients, thus better discriminating the risk for death compared to GFR calculated by means of MDRD-4 and/or CKD-EPI [14]. Beyond patients with CHF [32], the CG formula was able to predict mortality better than MDRD and CKD-EPI also in patients with acute decompensation of HF [33]. We demonstrated the better performance of CG compared to the other formulas in both acute and chronic HF. 

The most striking information related to our study was the demonstration of the good performance of the Donadio formula in predicting adverse events in HF patients. This is the first study dealing with the prognostic value of the Donadio formula in the setting of HF. The Donadio formula succeeded in combining bioimpedance analysis with the estimation of GFR, as previously demonstrated [17,18]. We also demonstrated that bioimpedance analysis may provide useful insights in HF patients: the evaluation of congestion by means of BIA and the association with further biomarkers may guide therapy, provide prognostic data, and predict length of stay [34,35,36,37,38]. The Donadio formula seemed to combine two fundamental aspects of HF pathogenesis: the evaluation of residual congestion/nutrition and the calculation of GFR, i.e., the assessment of cardio-renal syndrome. Indeed, there are no data about the prognostic value of eGFR as calculated by the Donadio formula in the setting of HF. We obtained an AUC of 0.72 with a cut-off < 51 mL/min/1.73 m^2^, the sensitivity and specificity set at 67% and 69%, respectively, while NPV was 89%. The Donadio equation was confirmed to act as an independent mortality risk factor in HF patients at multivariate Cox regression analysis, the Wald index being equal to 10.6.

These results are interesting. The formula created by Donadio et al. [17,18] allowed us to overcome some limitations related to creatinine-based equations. Donadio et al. [39] also observed amelioration in GFR measurements in mildly obese individuals. Nevertheless, further studies are needed to better address the daily application of the Donadio formula, and the contextual evaluation of congestion by means of BIA in patients with HF.

## 5. Limitations

The retrospective nature of this study may be considered as a limitation. Indeed, we collected data from a large population of patients with CHF or AHF, who were followed-up by cardiologists committed to daily management of HF.

The use of BIA in daily clinical practice is not widespread, and this issue may represent a possible limitation. Nevertheless, the subtle aim of this study was also the promotion of this specific technique: the simple, fast, costless application of bioimpedance analysis. The great amount of clinical and prognostic information derived from it may induce clinicians to adopt this technique for a comprehensive evaluation of patients with HF and, along with the Donadio formula, cardio-renal syndrome. 

The eGFR formulas had been all adjusted for body surface area (BSA). The literature provides concerns about this adjustment in clinical practice, as it may alter the correct estimation of eGFR, for example, in obese individuals [40,41]. Nevertheless, we adjusted all the formulas for BSA in order to make the results uniform to the aim of the study, which was to evaluate their prognostic impact in HF patients.

Equations may also underestimate eGFR in obese patients, above all when considering patients with severe forms of obesity (BMI > 35 Kg/m^2^) [42,43]. Indeed, no patient in our cohort showed very high BMI values, although the overall elevated BMI of the general population (mean value 28 ± 5 Kg/m^2^) might effectively impact on eGFR calculations. Although the Donadio equation seemed to perform well in mildly obese individuals [39], the role of obesity in GFR calculation remains challenging. Further studies are needed in order to overcome this bias.

The lack of information about proteinuria and albuminuria is a further matter of debate for this research, above all in relation to the percentage of the enrolled diabetic patients (24%). The impact of these parameters on cardio-renal syndrome has been previously established [44]. Albuminuria had been widely demonstrated to act as prognostic determinant in diabetic patients. The evaluation of albuminuria and proteinuria would add specific value to the multiparametric approach to HF patients for the comprehensive risk stratification of this category of patients.

The recent guidelines of the European Society of Cardiology on the management of HF [45] outlined the great positive influence of new antidiabetics in the prognosis of patients with HF. This study was performed before the introduction of glifozins in clinical practice. The confounding role of glifozins in the overall assessment of the prognosis of patients with HF would be an interesting starting point for further research.

A further limitation derives from the single center experience. Attempts to include further centers are ongoing, in the interest of demonstrating the validity of the results on larger populations.

## 6. Conclusions

The CG formula was confirmed to be an independent predictor of all-cause mortality in patients with AHF and CHF. The CG formula was superior to MDRD-4, Donadio, and CKD-EPI in the overall assessment of the prognosis of patients with HF. In addition, the application of BIA in a real-life cohort of patients with HF confirmed its prognostic value. Beyond the evaluation of congestion, BIA allows for the calculation of GFR and the prediction of adverse events related to kidney dysfunction in HF patients.

## Figures and Tables

**Figure 1 biomedicines-09-01307-f001:**
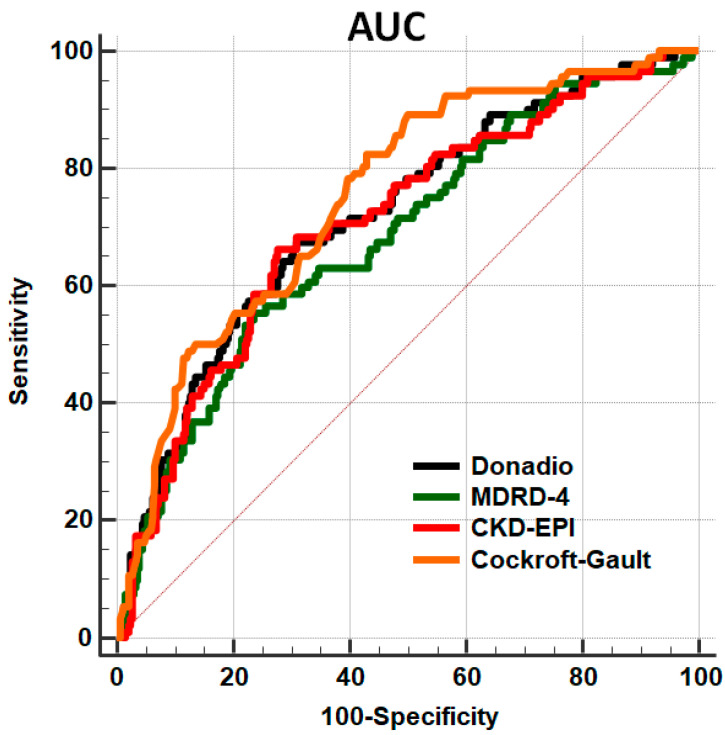
Area under the receiver operating characteristic curves (AUC) of the different eGFR formulas in predicting all-cause mortality. AUC derived from Cockroft-Gault formula was significantly higher than those from other formulas (*p* < 0.05 for all). Abbreviations: AUC: area under the curve; CKD-EPI: Chronic Kidney Disease Epidemiology Collaboration; MDRD-4: Modification of Diet in Renal Disease Study.

**Figure 2 biomedicines-09-01307-f002:**
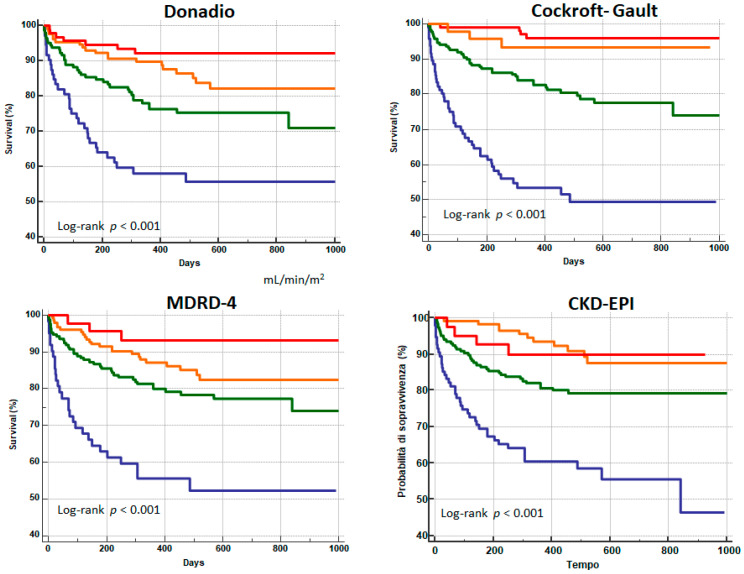
Kaplan-Meier survival curves relate to each formula for estimating glomerular filtration rate (GFR) at different stages of renal function: GFR: <30 mL/min/1.73 m^2^ (red); GFR between 30 and 59 mL/min/1.73 m^2^ (orange); GFR between 60 and 90 mL/min/1.73 m^2^ (green); and GFR > 90 mL/min/1.73 m^2^ (blue). Abbreviations: CKD-EPI: Chronic Kidney Disease Epidemiology Collaboration; MDRD-4: Modification of Diet in Renal Disease Study.

**Table 1 biomedicines-09-01307-t001:** Patient characteristics.

Clinical Characteristics	*n* = 436
Age, yrs	75 ± 11
Male, %	52
BMI, kg/m^2^	28 ± 5
NYHA I-II/III/IV, %	43/30/27
Peripheral oedema, %	30

Medical history, %	
Coronary artery disease	30
Diabetes	24
Atrial fibrillation	43
PM/ICD	18
AHF	42

LVEF, %	43 ± 12
Preserved LVEF, %	48
Mid-range LVEF, %	10
Reduced LVEF, %	42

Laboratory values	
BNP, pg/dL	510 (199–1100)
Hemoglobin, g/dL	13 ± 2
Uric acid, mg/dL	6.2 ± 2.1
BUN, mg/dL	30 ± 17
Creatinine, mg/dL	1.4 ± 0.9
Sodium, mmol/L	139 ± 4
Potassium, mmol/L	4.0 ± 0.6
Chloride, mmol/L	103 ± 5
Albumin, g/dL	3.3 ± 0.6

Bioelectrical Impedance Parameters	
Resistance/height, Ohm/m	309 ± 68
Reactance/height, Ohm/m	27 ± 8

Therapies, %	
Furosemide	69
Beta-blockers	50
ACE inhibitors/ARBs	60
MRAs	69
Digitalis	21
Ivabradine	5

Numbers are expressed as percentage or mean ± standard deviation. Abbreviations: ACE: angiotensin-converting enzyme; AHF: acute heart failure; ARB: angiotensin receptor blocker; BMI: body mass index; BNP: brain natriuretic peptide; BUN: blood urea nitrogen; ICD: implanted cardioverter/defibrillator; LVEF: left ventricular ejection fraction; MRAs mineralocorticoid receptor antagonists; NYHA: New York Heart Association; PM: pacemaker.

**Table 2 biomedicines-09-01307-t002:** Performance of eGFR equations in predicting all-cause mortality.

eGFR (mL/min/1.73 m^2^)	Non-Survivors (*n* = 72)	Survivors (*n* = 344)	AUC (95% CI)	Cut-Off	Sensitivity	Specificity	PPV	NPV	*p*
Donadio	45 ± 28	70 ± 35	0.72 (0.67–0.76)	<51	67	69	37	89	<0.0001
Cockroft-Gault	45 ± 20	61 ± 28	0.75 (0.71–0.79)	<50	82	57	33	92	<0.0001
MDRD-4	36 ± 20	58 ± 28	0.68 (0.64–0.73)	<43	54	76	38	86	<0.0001
CKD-EPI	38 ± 23	58 ± 29	0.71 (0.66–0.75)	<41	66	72	39	89	<0.0001

Abbreviations. AUC: area under the curve; CI: confidence interval; CKD-EPI: Chronic Kidney Disease Epidemiology Collaboration; eGFR: estimated glomerular filtration rate; MDRD-4: Modification of Diet in Renal Disease Study; NPV: negative predictive value; PPV: positive predictive value.

**Table 3 biomedicines-09-01307-t003:** Univariate Cox proportional hazards survival analyses.

	Univariate Cox Regression Analysis	
	HR (95% CI)	*p*
Donadio	0.976 (0.968–0.983)	<0.0001
Cockroft-Gault	0.962 (0.952–0.972)	<0.0001
MDRD-4	0.976 (0.967–0.984)	<0.0001
CKD-EPI	0.972 (0.963–0.981)	<0.0001
Coronary artery disease	0.962 (0.610–1.501)	ns
Diabetes	1.081 (0.681–1.722)	ns
Atrial fibrillation	1.422 (0.941–2.15)	ns
AHF vs. CHF	2.721 (1.782–4.12)	<0.0001
NYHA I-II/III/IV	1.823 (1.45–2.38)	<0.0001
LVEF, %	0.981 (0.971–1.102)	ns
BNP, pg/dL × 100	1.040 (1.030–1.051)	<0.0001
Hemoglobin, g/dL	0.777 (0.701–0.854)	<0.0001
Sodium, mmol/L	0.925 (0.945–10.67)	ns

Abbreviations. AHF: acute heart failure; BNP: brain natriuretic peptide; CHF: chronic heart failure; CI: confidence interval; CKD-EPI: Chronic Kidney Disease Epidemiology Collaboration; eGFR: estimated glomerular filtration rate; HR: hazard ratio; LVEF: left ventricle ejection fraction; MDRD-4: Modification of Diet in Renal Disease Study; ns = not significant; NYHA: New York Heart Association.

**Table 4 biomedicines-09-01307-t004:** Multivariate Cox proportional hazards survival analyses.

	Adjusted Cox Regression Analysis		
	HR (95% CI)	*p*	Wald

Donadio	0.985 (0.977–0.994)	=0.001	6.5
BNP, pg/dL × 100	1.03 (1.02–1.05)	=0.001	15.4

Cockroft-Gault	0.974 (0.962–0.985)	<0.001	17.3
BNP, pg/dL × 100	1.02 (1.01–10.4)	=0.002	9.4
NYHA I-II/III/IV	1.363 (1.020–1.816)	=0.036	4.4

MDRD-4	0.988 (0.979–0.998)	=0.03	4.7
BNP, pg/dL × 100	1.030 (1.020–1.040)	=0.0002	13.8
NYHA I-II/III/IV	1.336 (1.001–1.783)	<0.04	3.9
Hemoglobin, g/dL	0.886 (0.796–0.987)	=0.03	4.8

CKDEPI	0.984 (0.974–0.994)	=0.003	9.4
BNP, pg/dL × 100	1.03 (1.010–1.040)	=0.0004	12.7
NYHA I-II/III/IV	1.337 (1.001–1.785)	=0.04	17.3

Abbreviations. BNP: brain natriuretic peptide; CI: confidence interval; CKD-EPI: Chronic Kidney Disease Epidemiology Collaboration; HR: hazard ratio; MDRD-4: Modification of Diet in Renal Disease Study; NYHA: New York Heart Association.

## Data Availability

The data will be available on request by contacting the Corresponding Author.

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
