# Peer review of "The Prognostic Impact of Estimated Creatinine Clearance by Bioelectrical Impedance Analysis in Heart Failure: Comparison of Different eGFR Formulas"

_biomedicines, 2021, doi:10.3390/biomedicines9101307_

Round 1

Reviewer 1 Report

The paper by Scicchitano et al. deals with a retrospective study to assess the role of eGFR estimation with Donadio formula in predicting all cause mortality and to compare it with the other eGFR estimation formulas, in a cohort of heart failure patients. The study has a good design. The article is logically divided into sections and subsections. The methods are presented in sufficient detail and statistical analysis is not clearly reported in the right tables. The references cited are relevant and adequate. Discussion should be revised and enriched. The work has an average degree of novelty. English is fine, only minor spell check is needed.

Major comment:

1) One of the main issue is that evaluating kidney function by eGFR estimation with formulas in this specific population may be a lot biased. In fact, proteinuria and albuminuria (especially in diabetic subjects which represent the 24% of the entire population), have been both assessed of high impact in determining the cardiorenal outcome (and may also be mismatched with a decrease in glomerular filtration rate) (doi: 10.3390/diagnostics11020290). Moreover, newer antihyperglycemic drugs for diabetes are of high impact in reducing the all cause mortality and gliflozins are now, according to most recent guidelines (ESC 2021), fundamental in heart failure treatment, and are not reported in the analysis. Finally, other comorbidities may also affect outcome. All this should be reported in the discussion and limitations sections.

2) line 226-227: "The Donadio equation confirmed to act as an independent mortality risk factor...". I am not sure sure you can say that as univariate and multivariate analysis was not corrected for all risk factors such as the one aforementioned. Can you provide a table of univariate and multivariate analysis with all the risk factors not mentioned in table 3?  

Author Response

Reviewer #1

We thank this Reviewer for the constructive comments and suggestions. Furthermore, we would like to really thank him/her for his/her appreciation about our research. This is our point to point reply.

  1. One of the main issue is that evaluating kidney function by eGFR estimation with formulas in this specific population may be a lot biased. In fact, proteinuria and albuminuria (especially in diabetic subjects which represent the 24% of the entire population), have been both assessed of high impact in determining the cardiorenal outcome (and may also be mismatched with a decrease in glomerular filtration rate) (doi: 10.3390/diagnostics11020290). Moreover, newer antihyperglycemic drugs for diabetes are of high impact in reducing the all cause mortality and gliflozins are now, according to most recent guidelines (ESC 2021), fundamental in heart failure treatment, and are not reported in the analysis. Finally, other comorbidities may also affect outcome. All this should be reported in the discussion and limitations sections.

We would like to really thank the reviewer for these interesting insights. We updated the limitation section in order to include these comments. Thank you very much.

  1. line 226-227: "The Donadio equation confirmed to act as an independent mortality risk factor...". I am not sure sure you can say that as univariate and multivariate analysis was not corrected for all risk factors such as the one aforementioned. Can you provide a table of univariate and multivariate analysis with all the risk factors not mentioned in table 3? 

Thank you very much for your suggestions. We created two tables: one related to univariate and the second related to the multivariate analyses. Here we included all the unmentioned risk factors.

Reviewer 2 Report

I consider that the idea of this study, to evaluate the prognostic value of eGFR as assessed by Donadio formula, and to compare it to Cockcroft-Gault, MDRD-4 and CKD-EPI formulas in HF patients is very interesting, considering that monitoring renal function is one of the mainstays in the general management of patients with heart failure. The article is well-written and comprehensive. The study is correctly designed and the results are clearly and transparently presented. The set goals correspond to the conclusions. Even though this study is retrospective and included patients from a single-center, I consider that the findings are interesting and that the results obtained can make significant contributions to further large studies.

However, given that there are other recent studies that evaluated the prognostic impact of bioelectrical impedance analysis in patients with heart failure, I recommend emphasizing the original elements of the study.

Author Response

Reviewer #2

We thank this Reviewer for her/his useful suggestions. We sincerely appreciate his/her comments on our work. This is our point-to-point reply:

  1. I consider that the idea of this study, to evaluate the prognostic value of eGFR as assessed by Donadio formula, and to compare it to Cockcroft-Gault, MDRD-4 and CKD-EPI formulas in HF patients is very interesting, considering that monitoring renal function is one of the mainstays in the general management of patients with heart failure. The article is well-written and comprehensive. The study is correctly designed and the results are clearly and transparently presented. The set goals correspond to the conclusions. Even though this study is retrospective and included patients from a single-center, I consider that the findings are interesting and that the results obtained can make significant contributions to further large studies.

However, given that there are other recent studies that evaluated the prognostic impact of bioelectrical impedance analysis in patients with heart failure, I recommend emphasizing the original elements of the study.

We really thank the reviewer for the appreciation of our work and his/her useful insight. We updated the innovative aspects of our research in the discussion section in order to better outline the novelty of our findings.

Reviewer 3 Report

Scicchitano and coworkers presented an interesting study: The prognostic impact of estimated creatinine clearance by bioelectrical impedance analysis in heart failure: comparison with Cockcroft-gault, MDRD-4 and CKD-EPI formula.

The authors addressed an interested topic with specific population of patients, presented an elegant study, highlighted the strengths of their study, stressed limitations and presented some interesting results and conclusions.

Closer look has raised some comments:

  1.     The title may be shorter (“comparison of different eGFR formulas”).
  2.     The C&G formula is originally represented by a “clearance” unit - ml/min. This fact needs to be discussed more in detail as the authors calculated the eGFR of all formulas at a BSA 1.73m2.
  3.     Also, the impact of the marginal BMI (low and high) on BSA should be discussed. 

The authors should accept and discuss these comments.

Author Response

Reviewer #3

We thank this Reviewer for her/his useful suggestions. We sincerely appreciate his/her comments on our work. This is our point-to-point reply:

  1. The title may be shorter (“comparison of different eGFR formulas”).

Thank you very much for the suggestion. We accordingly revised the title.

  1. The C&G formula is originally represented by a “clearance” unit - ml/min. This fact needs to be discussed more in detail as the authors calculated the eGFR of all formulas at a BSA 1.73m2.

We really thank the reviewer for this interesting insight. We updated the limitation section by discussing such important issue.

  1. Also, the impact of the marginal BMI (low and high) on BSA should be discussed.

Once again we would like to really thank the reviewer for this updated and constructive comment. We discussed such a point in the limitation section of the paper in order to enforce the insight from the reviewer.

Round 2

Reviewer 1 Report

The authors answered to all my request and the manuscript improved. The paper, in my opinion, can  now be further processed for publication.